# COVER FILTRATION AND STABLE PATHS IN MAPPER

## ABSTRACT

The contributions of this paper are two-fold. We define a new filtration called the *cover filtration* built from a single cover based on a generalized Jaccard distance. We provide stability results for the cover filtration, and show its equivalence to the Čech filtration under certain settings. We then develop a language and theory for stable paths within this filtration, inspired by ideas of persistent homology. This framework can be used to develop new learning representations in applications where an obvious metric may not be defined but a cover is readily available. We demonstrate the utility of our framework as applied to recommendation systems and explainable machine learning.

We demonstrate a new perspective for modeling recommendation system data sets that does not require defining a bespoke metric. As an application, we find that stable paths identified by our framework in a movies data set represent sequences of movies constituting a gentle transition and ordering from one genre to another.

For explainable machine learning, we present stable paths between subpopulations in the *mapper* of a model as explanations. Our framework provides an alternative way to build a filtration from a single mapper, and explore stable paths in it. For illustration, we build a mapper from a supervised machine learning model trained on the FashionMNIST data set. Stable paths in the cover filtration provide improved explanations of relationships between subpopulations of images.

**Keywords:** cover and nerve, Jaccard distance, stable paths in filtration, Mapper, recommender systems, explainable machine learning.

## 1 INTRODUCTION AND MOTIVATION

The need to rigorously seed a solution with a notion of stability in topological data analysis (TDA) has been addressed primarily using topological persistence (Carlsson, 2009; Ghrist, 2008). Persistence arises when we work with a sequence of objects built on a data set, a *filtration*, rather than with a single object. One line of focus of this work has been on estimating the homology of the data set. This typically manifests itself as examining the persistent homology represented as a diagram or barcode, with interpretations of zeroth and first homology as capturing significant clusters and holes, respectively (Adams & Carlsson, 2009; Edelsbrunner et al., 2002; Edelsbrunner & Harer, 2009; Zomorodian, 2005). In practice it is not always clear how to interpret higher dimensional homology (even holes might not make obvious sense in certain cases). A growing focus is to use persistence diagrams as a form of feature engineering to help compare different data sets rather than interpret individual homology groups (Adams et al., 2017; Chazal et al., 2009; Turner et al., 2014).

The implicit assumption in most such TDA applications is that the data is endowed with a natural metric, e.g., points exist in a high-dimensional space or pairwise distances are available. In certain applications, it is also not clear how one could assign a meaningful metric. For example, memberships of people in groups of interest is captured simply as sets specifying who belongs in each group. An instance of such data is that of recommendation systems, e.g., as used in Netflix to recommend movies to the customer. Graph based recommendation systems have been an area of recent research. Usually these systems are modeled as a bipartite graph with one set of nodes representing recommendees and the other representing recommendations. In practice, these systems are augmented in bespoke ways to accommodate whichever type of data is available. It is highly desirable to analyze the structure directly using the membership information.

Another distinct TDA approach for structure discovery and visualization of high-dimensional data is based on a construction called *Mapper* (Singh et al., 2007). Defined as a dual construction to a *cover* of the data (see Figure 1), Mapper has found increasing use in diverse applications in the past several

years (Lum et al., 2013). Attention has recently focused on interpreting parts of the 1-skeleton of the Mapper, which is a simplicial complex, as significant features of the data. Paths, flares, and cycles have been investigated in this context (Li et al., 2015; Nicolau et al., 2011; Torres et al., 2016). The framework of persistence has been applied to this construction to define a *multi-scale* Mapper, which permits one to derive results on stability of such features (Dey et al., 2016). At the same time, the associated computational framework remains unwieldy and still most applications base their interpretations on a single Mapper object.

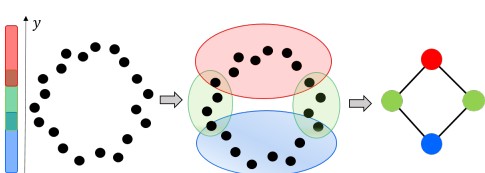

Figure 1: Mapper constructed on a noisy set of points sampled from a circle.

We illustrate the standard Mapper construction in Figure 1. We start with overlapping intervals covering values of a parameter, e.g., height of the points sampled from a circle. We then cluster the data points falling in each interval, and represent each cluster by a vertex. If two clusters share data points, we add an edge connecting the corresponding vertices. If three clusters share data points, we add the triangle, and so on. The Mapper could present a highly sparse representation of the data set that still captures its structure—the large number of points sampled from the circle is represented by just four vertices and four edges here. More generally, we consider higher dimensional intervals covering a subspace of $\mathbb{R}^d$. But in recommendation systems, the cover is just a collection of abstract sets providing membership info (rather than intervals over the range of function values). Could we define a topological construction on such abstract covers that still reveals the topology of the dataset?

We could study paths in this construction, but as the topological constructions are noisy, we would want to define a notion of stability for such paths. With this goal in mind, could we define a *filtration* from the abstract cover? But unlike in the setting of, e.g., multiscale Mapper (Dey et al., 2016), we do not have a sequence of covers (called a tower of covers)—we want to work with a *single* cover. How do we define a filtration on a single abstract cover? Could we prove stability results for such a filtration? Finally, could we demonstrate the usefulness of our construction on real data?

## 1.1 OUR CONTRIBUTIONS

We introduce a new type of filtration defined on a single abstract cover. Termed *cover filtration*, our construction uses Steinhaus distances between elements of the cover. We generalize the Steinhaus distance between two elements to those of multiple elements in the cover, and define a filtration on a single cover using the generalized Steinhaus distance as the filtration index. Working with a bottleneck distance on covers, we show a stability result on the cover filtration—the cover filtrations of two covers are $\alpha/m$ interleaved, where $\alpha$ is a bound on the bottleneck distance between the covers and $m$ is the cardinality of the smallest element in either cover (see Theorem 3.5). We conjecture that in Euclidean space, the cover filtration is isomorphic to the standard Čech filtration built on the data set. We prove the conjecture holds in dimension 1 and independently that the Vietoris-Rips filtration completely determines the cover filtration in arbitrary dimensions (see Section 4).

This filtration is quite general, and enables the computation of persistent homology for data sets without requiring strong assumptions. With real life applications in mind, we study paths in the 1-skeleton of our construction. Paths provide intu-

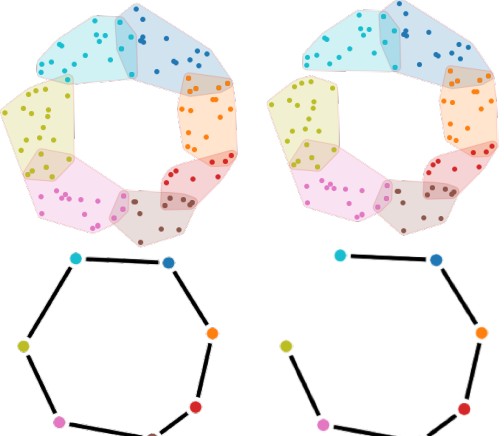

Figure 2: A cover with 7 elements, and the corresponding nerve (left column). The cyan and green vertices are connected by a single edge. But this edge is generated by a *single* point in the intersection of the cyan and green cover elements. Removing this point from the data set gives the cover and nerve shown in the right column. The path from cyan to green node now has six edges.

itive explanations of the relationships between the objects that the terminal vertices represent. Our perspective of path analysis is that shortest may not be more descriptive—see Figure 2 for an illustration. Instead, we define a notion of *stability* of paths in the cover filtration. Under this notion, a stable path is analogous to a highly persistent feature as identified by persistent homology.

We demonstrate the utility of stable paths in cover filtrations on two real life applications: a problem in movie recommendation system and Mapper. We first show how recommendation systems can be modeled using the cover filtration, and then show how stable paths within this filtration suggest a sequence of movies that represent a "smooth" transition from one genre to another (Appendix C.1). We then define an extension of the traditional Mapper (Singh et al., 2007) termed the *Steinhaus Mapper Filtration*, and show how stable paths within this filtration can provide valuable explanations of populations in the Mapper, focusing on the case of explainable machine learning (Appendix C.2).

## 1.2 RELATED WORK

Cavanna & Sheehy (2017) developed theory for a *cover filtration*, built from a cover of a filtered simplicial complex. But we work from more general covers of arbitrary spaces.

We are inspired by similar goals as those of Dey et al. (2016) and Carrière et al. (2018), who addressed the question of stability in the Mapper construction. Our goal is to provide some consistency, and thus interpretability, to the Mapper. We incorporate ideas of persistence in a different manner into our construction using a *single* cover, which considerably reduces the effort to generate results.

While stability of persistence diagrams is a well studied topic (Edelsbrunner & Harer, 2009), we may not get stability to simplices and cycles that generate the persistence homology classes (Bendich et al., 2019). In contrast, our definition of path stability (Section 5) aligns exactly with our result on stability for cover filtrations (Section 3). We believe this is a unique feature of our method and its stability result—paths are automatically stable with respect to perturbations of input data.

The multi-scale Mapper defined by Dey et al. (2016) builds a filtration on the Mapper by varying the parameters of a cover. This construction yields nice stability properties, but is unwieldy in practice and difficult to interpret. Carrière et al. develop ways based on extended persistence to automatically select a single cover that best captures the topology of the data (Carrière et al., 2018), producing one final Mapper that is easy to interpret

Methods have been developed to track populations within the Mapper by identifying interesting paths (Kalyanaraman et al., 2019) and interesting flares (Kamruzzaman et al., 2018). Interesting paths maximize an interestingness score, and are manifested in the Mapper as long paths that track particular populations that show trending behavior. Flares capture subpopulations that diverge, i.e., show branching behavior. In our context, we are interested in shorter paths, under the assumption that they provide the most succinct explanations for relationships between subpopulations.

Our work is similar to that of Parthasarathy et al. (2017) in that they use the Jaccard Index of an observed graph to estimate the geodesic distance of the underlying graph. We take an approach more akin to persistence and make fewer assumptions about underlying data. Hence we are unable to make rigorous estimates of distances and instead provide many possible representative paths.

S-paths defined by Purvine et al. (2018) are similar to stable paths when we realize that covers can be modeled as hypergraphs, and vice versa. Stable paths incorporate the size of each cover elements (or hyperedges), normalizing the weights by relative size. This perspective allows us to compare different parts of the resulting structure which may have wildly difference sizes of covers. In this context, a large overlap of small elements is considered more meaningful than a proportionally small intersection of large elements.

## 2 COVER FILTRATIONS

We introduce the notions of distance on covers required to construct our filtrations and then provide the general definition of the cover filtration.

We begin with the definition of Steinhaus distance, a generalization of the standard Jaccard distance between two sets, and further generalize it to an arbitrary collection of subsets of a cover. These generalizations take a measure $\mu$ and assumes that all sets are taken modulo differences by sets of measure 0. We will take $\mu$ as arbitrary unless otherwise stated.

**Definition 2.1** (Steinhaus Distance (Marczewski & Steinhaus, 1958)). Given a measure $\mu$, the *Steinhaus distance* between two sets $A, B$ is

$$d_{St}(A, B) = 1 - \frac{\mu(A \cap B)}{\mu(A \cup B)} = \frac{\mu(A \cup B) - \mu(A \cap B)}{\mu(A \cup B)}.$$

This distance is also bounded on $[0, 1]$, i.e., two sets have Steinhaus distance $0$ when they are equal (as sets differing by measure $0$ are identified) and distance $1$ when they do not intersect.

We extend this distance from an operator on a pair of elements to an operator on a set of elements.

**Definition 2.2** (Generalized Steinhaus distance). Define the *generalized Steinhaus distance* of a collection of sets $\{U_i\}$ as

$$d_{St}(\{U_i\}) = 1 - \frac{\mu(\bigcap U_i)}{\mu(\bigcup U_i)}.$$

Much of our results that consider paths used the standard version of Jaccard distance for finite sets. But we require the Steinhaus distance for general measures to present equivalence results to Čech and VR filtrations (Section 4). Hence we present all results for the generalized Steinhaus distance.

We make use of this generalized distance to associate birth times to simplices in a nerve. Given a cover, we define the cover filtration as the filtration induced from sublevel sets of the generalized Steinhaus distance function. In other words, consider a cover of the space and the nerve of this cover. For each simplex in the nerve, we assign as birth time the value of its Steinhaus distance. This filtration captures information about similarity of cover elements, and the overall structure of the cover.

**Definition 2.3** (Nerve). A *nerve* of a cover $\mathcal{U} = \{U_i\}_{i \in C}$ is an abstract simplicial complex defined such that each subset $\{U_j\}_{j \in J} \subseteq \mathcal{U}$, i.e., with $J \subseteq C$, defines a simplex if $\bigcap_{j \in J} \{U_j\} \neq \emptyset$. In this construction, each cover element $U_i \in \mathcal{U}$ defines a vertex.

**Definition 2.4** (Steinhaus Nerve). The *Steinhaus nerve* of a cover $\mathcal{U}$, denoted $\mathrm{Nrv}_{St}(\mathcal{U})$, is defined as the nerve of $\mathcal{U}$ with each simplex assigned their generalized Steinhaus distance as weight:

$$w_\sigma = d_{St}(\{U_i \mid i \in \sigma\}) \ \forall \sigma \in \mathcal{U}.$$

Note that $w_\sigma < 1$ by definition for every simplex $\sigma \in \mathcal{U}$. We will use $\mathrm{Nrv}_{St}$ when the cover $\mathcal{U}$ is evident from context. This can be thought of as a weighted nerve, but the weighting scheme satisfies the conditions of a filtration (see Page 14 for the proof).

**Theorem 2.5.** *The Steinhaus nerve of a cover $\mathcal{U}$ is a filtered simplicial complex.*

Following this result, we refer to the construction as the Steinhaus *filtration*. The only cover filtrations we use in this paper is Steinhaus filtrations, so we will use the two terms interchangeably.

We could study an adaptation of cover filtration to an analog of the VR complex by building a weighted clique rank filtration from the 1-skeleton of the cover filtration (Petri et al., 2013). This adaptation drastically reduces the number of intersection and union checks required for the construction. The *weight rank clique filtration* is a way of generating a flag filtration from a weighted graph (Petri et al., 2013). We can apply this technique to build a VR analog of the cover filtration.

## 3   STABILITY

We consider notions of stability in the cover filtration with respect to changes in the cover. We first modify the standard edit distance to define a bottleneck distance on the space of covers of a finite set that have the same cardinality. Under this setting, we show that the cover filtration is interleaved with respect to this distance.

**Definition 3.1** (Bottleneck metric on covers). Let $\mathcal{U}$ and $\mathcal{V}$ be two finite covers of finite set $X$ with same cardinality, and let $\mathcal{M}(\mathcal{U}, \mathcal{V})$ be the set of all possible matchings between them. Let $\triangle$ denote the symmetric difference. Then the bottleneck distance $d_B(\cdot, \cdot)$ between two covers is defined as

$$d_B(\mathcal{U}, \mathcal{V}) = \min_{M \in \mathcal{M}(\mathcal{U}, \mathcal{V})} \left\{ \max_{(U, V) \in M} \mu(U \triangle V) \right\}.$$

We can verify that $d_B$ is indeed a metric (see Page 14 for the proof).

**Proposition 3.2.** *Let $\mathcal{U}, \mathcal{V}$, and $\mathcal{W}$ be finite covers of a finite set $X$ with equal cardinalities, and let $d_B$ be the bottleneck distance between any pair of these covers (Definition 3.1). Then $d_B$ is a metric.*

We now present two somewhat technical lemmas, which we subsequently employ in the proof of the main stability result. See Page 14 for proof of Corollary 3.4.

**Lemma 3.3.** *Let $a, b, c, d, e, f$ be real numbers with $a < b$, $c + d = e + f$, $|b| > |d|$, and $|b| > |f|$. Then we have that $(a-c)/(b+d) > (a-e)/(b+f)$ when $c < e$. In words, for a weight $\alpha = c+d$ that we can distribute between decreasing the numerator of a proper fraction and increasing its denominator, the greatest decrease comes from decreasing the numerator by the entire weight.*

*Proof.* We will show that, with the given conditions, $\dfrac{a-c}{b+d} - \dfrac{a-e}{b+f} > 0$.

$$
\begin{aligned}
\frac{a-c}{b+d} - \frac{a-e}{b+f} &= \frac{(a-c)(b+f) - (a-e)(b+d)}{(b+d)(b+f)} \\
&= \frac{(a-c)(b+c+d-e) - (a-e)(b+e+f-c)}{(b+d)(b+f)}, \text{ since } f = c+d-e,\ d = e+f-c. \\[1em]
&= \frac{ab+ac+ad-ae-cb-c^2-cd+ce-ab-ae-af+ac+eb+e^2+ef-ce}{(b+d)(b+f)} \\
&= \frac{2ac-2ae+ad-af+be-bc+e^2-c^2+ef-cd}{(b+d)(b+f)} \\
&= \frac{-2a(e-c)+a(d-f)+b(e-c)+(e+c)(e-c)+ef-cd}{(b+d)(b+f)} \\
&= \frac{-2a(e-c)+a(e-c)+b(e-c)+(e+c)(d-f)+ef-cd}{(b+d)(b+f)}, \\
&\quad \text{since } e-c = d-f. \\
&= \frac{(-a+b)(e-c)+ed-ef+cd-cf+ef-cd}{(b+d)(b+f)} = \frac{(-a+b)(e-c)+ed-cf}{(b+d)(b+f)}\,.
\end{aligned}
$$

Since $c < e$, $d > f$, so $ed > cf$ and $e-c > 0$. Also, since $a/b$ is proper, $a < b$, so $-a+b > 0$. Thus $(-a+b)(e-c)+(ed-cf) > 0$. Since $b+d, b+f > 0$, we get that $\dfrac{(-a+b)(e-c)+ed-cf}{(b+d)(b+f)} > 0$, which in turn shows that $\dfrac{a-c}{b+d} > \dfrac{a-e}{b+f}$, as desired. $\square$

**Corollary 3.4.** *Similar to Lemma 3.3, the greatest increase possible in such a scenario comes from assigning the negative of the total weight to the numerator.*

We now present a theorem that provides basic stability guarantees for the constructed filtration, assuming that each element is not too small.

**Theorem 3.5.** *Suppose that $\mathcal{U} = \{U_i\}$ and $\mathcal{V} = \{V_j\}$ are two covers of $X$ with $|\mathcal{U}| = |\mathcal{V}|$ such that $d_B(\mathcal{U}, \mathcal{V}) \leq \alpha$. Given $m = \min\{\min_{U \in \mathcal{U}} \mu(U), \min_{V \in \mathcal{V}} \mu(V)\}$, $\mathrm{Nrv}_{St}(\mathcal{U})$ and $\mathrm{Nrv}_{St}(\mathcal{V})$ are $\alpha/m$ interleaved filtrations.*

*Proof.* Given two covers $\mathcal{U} = \{U_i\}$ and $\mathcal{V} = \{V_j\}$, we use the notation that $U_i$ and $V_i$, for generic indices $i$ and $j$, are paired in a matching that minimizes the bottleneck distance between the two covers. We assume that the bottleneck distance is $\alpha$, a positive integer.

We consider the following question: what is the largest change in generalized Steinhaus distance possible between a collection $U_I$ and $V_J$, where index sets $I$ and $J$ are paired elementwise in a matching. To answer this question, we keep $U_I$ fixed and consider how large a difference in generalized Steinhaus distance we can achieve by taking the symmetric difference with a set $S_i$ with

measure up to $\alpha$ for each $U_i$. That is, we want to maximize change in $\mu(\cap U_I)/\mu(\cup U_I)$. To get the maximum increase in $\mu(\cap U_I)/\mu(\cup U_I)$, we must increase the numerator and/or decrease the denominator. Likewise, we must decrease the numerator and/or increase the denominator to get maximum decrease.

First, we note that $S_i$ can be partitioned into two sets: $s_{i,1} = U_i \cap S_i$, and $s_{i,2} = S_i \setminus s_{i,1}$, those not in $U_i \cap S_i$. $\mu(s_{i,1} + s_{i,2}) = \mu(S_i) \leq \alpha$. Replacing $U_i$ with $U_i \setminus s_{i,1}$ cannot increase the size of the intersection or union, but it can decrease the size of the intersection or union by up to $\mu(s_{i,1})$. Likewise, replacing $U_i$ with $U_i \cup s_{i,2}$ cannot decrease the size of the intersection or union, but could increase the size of the intersection or union by up to $\mu(s_{i,2})$.

The greatest possible change would occur if it were possible to select an $S_i$ for each $U_i$ in the cover such that replacing each $U_i$ in turn with $U_i \triangle S_i$ increases or decreases the numerator with $\mu(s_{i,1})$ or $\mu(s_{i,2})$, respectively, and does the opposite to the size of the denominator by $\mu(s_{i,2})$ or $\mu(s_{i,1})$, again respectively. Since $\mu(s_{i,1} + s_{i,2}) = \mu(S_i) \leq \alpha$, it follows that each change has a weight of at most $\alpha$ which it could throw into increasing or decreasing the size of the intersection and doing the opposite to the size of the union.

Lemma 3.3 and Corollary 3.4 imply that the maximum possible change in those situations will be achieved when all weight is directed toward increasing or decreasing the size of the intersection, since $\mu(\cap U_I)/\mu(\cup U_I)$ must be between 0 and 1. As we want to bound the possible change in the Steinhaus distance, it will suffice to use the observation that

$$1 - \frac{\mu(\cap V_I) + S\alpha}{\mu(\cup V_I)} \leq 1 - \frac{\mu(\cap U_I)}{\mu(\cup U_I)} \leq 1 - \frac{\mu(\cap V_I) - S\alpha}{\mu(\cup V_I)}$$

to obtain bounds on the change of Steinhaus distance between covers with maximum bottleneck distance of $\alpha$ and with $S$ as the cover cardinality.

Then we have

$$1 - \frac{\mu(\cap V_I) + S\alpha}{\mu(\cup V_I)} \leq 1 - \frac{\mu(\cap U_I)}{\mu(\cup U_I)}$$

$$\Rightarrow 1 - \frac{\mu(\cap V_I)}{\mu(\cup V_I)} - \frac{S\alpha}{\mu(\cup V_I)} \leq d_{St}(U_I)$$

$$\Rightarrow 1 - \frac{\mu(\cap V_I)}{\mu(\cup V_I)} - \frac{S\alpha}{Sm} \leq d_{St}(U_I), \text{ where } m \text{ is the size of the smallest set, since}$$

$$1 - \frac{\mu(\cap V_I)}{\mu(\cup V_I)} - \frac{S\alpha}{Sm} \leq 1 - \frac{\mu(\cap V_I)}{\mu(\cup V_I)} - \frac{S\alpha}{\mu(\cup V_I)},$$

$$\Rightarrow d_{St}(V_I) - \frac{\alpha}{m} \leq d_{St}(U_I).$$

Similarly, since

$$1 - \frac{\mu(\cap V_I) - S\alpha}{\mu(\cup V_I)} \geq 1 - \frac{\mu(\cap U_I)}{\mu(\cup U_I)}, \quad \text{we get} \quad d_{St}(V_I) + \frac{\alpha}{m} \geq d_{St}(U_I).$$

Hence $d_{St}(V_I) - \frac{\alpha}{m} \leq d_{St}(U_I) \leq d_{St}(V_I) + \frac{\alpha}{m}$, giving that $\mathcal{U}$ and $\mathcal{V}$ are $\alpha/m$ interleaved. $\quad\square$

*Remark* 3.6. Consider the case when $|\mathcal{U}| \neq |\mathcal{V}|$. Assume without loss of generality that $|\mathcal{U}| > |\mathcal{V}|$. Then there is a vertex $v \in \text{Nrv}_{St}(\mathcal{U})$ that is not present in $\text{Nrv}_{St}(\mathcal{V})$. Hence $\text{Nrv}_{St}(\mathcal{U})$ and $\text{Nrv}_{St}(\mathcal{V})$ cannot be interleaved in the current setting. We need to first generalize matchings and the bottleneck distance to allow covers with unequal cardinalities.

## 4 EQUIVALENCE

To situate the cover filtration, we wish to show that it is isomorphic to the Čech and VR filtrations under certain conditions. We conjecture that the Čech filtration on a finite set of points, i.e., the nerve of balls with radius $r$ around each point and over a sequence of $r$, and the Steinhaus Nerve constructed from the terminal cover of the Čech filtration are isomorphic.

More precisely, the insertion order of simplices is equivalent between the two cases, and there exists a continuous bijection between insertion times of the Steinhaus Nerve and insertion times of the Čech filtration. We prove this result for $n = 1$, i.e., when $X$ is drawn from the real line. We also provide experimental evidence for the 1-skeleton equivalence, and prove one direction of this equivalence (that the VR filtration completely determines the cover filtration).

Let $\check{C}_r(X)$ be the cover of $X$ by balls of radius $r$ centered on points in $X$. The Čech complex is nerve of this cover. The Čech filtration is the sequence of simplicial complexes for all values of $r$.

**Conjecture 4.1** (Čech equivalence). *Given a finite data set $X \subset \mathbb{R}^n$ and some radius $R > diam(X)$ the Čech filtration constructed from $X$ is isomorphic to the the cover filtration on $X$ constructed from $\check{C}_R(X)$, given the Lebesgue (i.e. volume) measure.*

A comprehensive proof for arbitrary order of intersections and arbitrary dimension is incomplete. We provide the proof for the case of $n$-skeleton in 1-dimension and provide a mapping from the Čech filtration to cover filtration for the 1-skeleton in arbitrary dimension.

*Proof.* We will define $\check{C}(\{v_i\})$ as the birth radius of the simplex defined by the set $\{v_i\}$. This can be computed as $\check{C}(\{v_i\}) = (\max_i(v_i) - \min_i(v_i))/2$ since in a 1-dimensional space the associated simplex is born precisely when the balls around the two outermost points intersect.

Let $\{V_i\}$ be the set of balls of radius $R$ centered on the set $\{v_i\}$. Recall that we are using the Steinhaus distance for Lebesgue measure, so $\mu$ computes volume here. Then the generalized Steinhaus distance for those balls is given by $d_{St}(\{V_i\}) = 1 - \frac{\min(v_i + R) - \max(v_i - R)}{\max(v_i + R) - \min(v_i - R)}$, since the mutual intersection of all of the balls in this one dimensional space is the interval bounded by the minimum right endpoint of all the balls and the maximum of all left endpoints of all the balls and the union of all the balls has the minimum left endpoint amongst left endpoints and maximum right endpoint amongst right endpoints.

$$d_{St}(\{V_i\}) = 1 - \frac{\min(v_i) - \max(v_i) + 2R}{\max(v_i) - \min(v_i) + 2R} = 1 - \frac{-2\check{C}(\{v_i\}) + 2R}{2\check{C}(\{v_i\}) + 2R} = 1 - \frac{R - \check{C}(\{v_i\})}{R + \check{C}(\{v_i\})}.$$

Solving for $\check{C}(\{v_i\})$, we get $\check{C}(\{v_i\}) = \frac{R d_{St}(\{V_i\})}{2 - d_{St}(\{V_i\})}$, establishing a bijection between birth times.

Now since $R$ is the radius of $X$ and $v_i \subseteq X, 0 \leq \check{C}(\{v_i\}) \leq R$. Also, $\frac{R-x}{R+x}$ decreases monotonically over the range $x \in [0, R]$. Thus $1 - \frac{R-x}{R+x}$ increases monotonically on $x \in [0, R]$. Thus if we order the subsets of of $X$ by increasing birth radius $(s_1, s_2, \cdots, s_n)$, $(d_{St}(s_1), d_{St}(s_2), \cdots, d_{St}(s_n))$ is also in increasing order. Thus the two filtrations are isomorphic. $\square$

We now address the case of 1-skeleton of the Čech and cover filtrations in arbitrary dimension. It is clear that if the 1-skeletons are isomorphic, then the cover filtration is isomorphic to the VR filtration. We prove one direction of this isomorphism (see Page 15 for the proof).

**Lemma 4.2.** *The VR filtration completely determines the cover filtration in arbitrary dimensions.*

Hence one can derive the cover filtration from the Vietoris-Rips filtration. In Appendix B, we present some experimental evidence suggesting that the 1-skeleton of the Steinhaus filtration and the Vietoris-Rips filtration are isomorphic.

# 5 STABLE PATHS

We develop a theory of stable paths in the 1-skeleton of a cover filtration. We provide an algorithm to find a maximally stable path from one vertex to another, with Steinhaus distances as edge weights. Note that a maximally stable path might not be a shortest path in terms of number of edges. Conversely, a shortest path might not be highly stable. Since the two objectives are at odds with each other, we provide an algorithm to identify a family of shortest paths as we vary the stability level, similar in a loose sense to computing persistent homology.

We were studying shortest paths in a Mapper constructed on a machine learning model as ways to illustrate the relations between the data as identified by the model. In this context, shortest paths found could have low Steinhaus distance, and thus could be considered noise. This motivated our

desire to find stable paths, as they would intuitively be most representative of the data set and stable with respect to changing parameters in Mapper or changing data.

**Definition 5.1** ($\rho$-Stable Path). Given a Steinhaus distance $\rho$, a path $P$ is defined to be *$\rho$-stable* if

$$\max\{d_{St}(e)|e \in P\} \leq \rho.$$

In other words, the largest edge weight (Steinhaus distance) along the path is at most $\rho$. Note that a $\rho_1$-stable path is also $\rho_2$-stable for any $\rho_2 \geq \rho_1$. Also, a $\rho_1$-stable path $P_1$ is more stable than a $\rho_2$-stable path $P_2$ when $\rho_1 < \rho_2$. Hence we have a higher confidence that edges in $P_1$ do exist (are *not* due to noise) than those in $P_2$. We now define maximally stable paths between a pair of vertices.

**Definition 5.2** (Maximally Stable Path). Given a pair of vertices $s$ and $t$, a *maximally stable s-t path* is a $\rho$-stable path between $s$ and $t$ for the smallest value of $\rho$. If there are multiple $s$-$t$ paths at the same minimum $\rho$ value, a shortest path among them is defined as a maximally stable path.

Figure 3: Algorithm to identify the Pareto frontier between shortest and maximally stable paths.

```
Input: 1-skeleton G of cover filtration
       and vertices s,t

set LIST = [∅,∅] // stores [P,ρ] pairs
while s,t are connected in G
  compute shortest path P between s and t
  find ρ = max{d_St(e)|e ∈ P}

  if LIST has no pair [P',ρ'] : |P| = |P'|
    add [P,ρ] to LIST
  else if ρ < ρ' for [P',ρ'] ∈LIST : |P| = |P'|
    replace [P',ρ'] with [P,ρ] in LIST

  remove all edges e from G with d_St(e) ≥ ρ

Return: LIST
```

The problem of finding the maximally stable $s$-$t$ path can be solved as a minimax path problem on an undirected graph, which can solved efficiently using, e.g., range minimum queries (Demaine et al., 2009).

We are then left with two paths between vertices $s$ and $t$, the shortest and the maximally stable. The shortest path is not necessarily stable and the stable path is not necessarily short. As these two notions, stable and short, are at odds with each other, we are interested in computing the entire Pareto frontier between the short and stable path. We present an algorithm to identify the Pareto frontier in Figure 3, and a visualization of the output from this algorithm in Figure 4.

We repeatedly compute the shortest path, while essentially sweeping over the Steinhaus Distance. This process results in a Pareto frontier balancing the shortest paths with the stability of those paths. The blue points in Figure 4 are on the Pareto frontier, while the orange points are the pairs $[P', \rho']$ that get replaced from the LIST in the course of the algorithm. We then visualize the paths on the Pareto frontier in Figure 5. Continuing our analogy to persistence, the path corresponding to a point on the Pareto frontier which sees a steep rise to the left is considered highly persistent, e.g., the path with length 21 on the frontier.

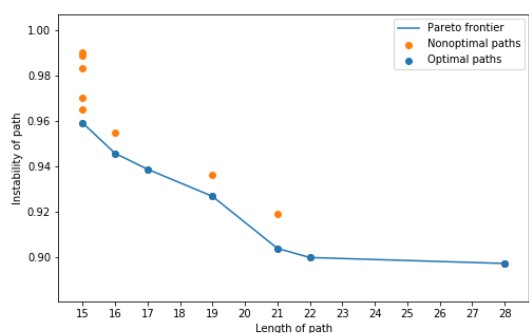

Figure 4: Pareto frontier between length and stability of path.

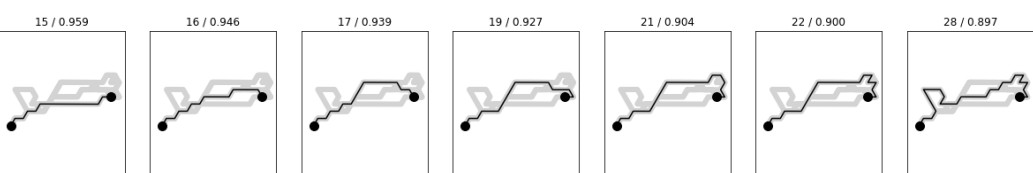

Figure 5: Visualization of paths on the Pareto frontier in Figure 4. Length$/\rho$ listed on top.

## 6 APPLICATIONS

We apply the cover filtration and stable paths to two applications, recommendation systems and Mapper. We first show how recommendation systems can be modeled using the cover filtration and then show how stable paths within this filtration can answer the question *what movies should I show my friend first, to wean them into my favorite (but potentially weird) movie?* We then define an extension of the traditional Mapper (Singh et al., 2007; Carrière & Oudot, 2017) called the Steinhaus Mapper Filtration, and show how stable paths within this filtration can provide valuable explanations of populations in the Mapper. As a direct illustration, we focus on the case of explainable machine learning, where the Mapper is constructed with a supervised machine learning model as the filter function, and address the question *what can we learn about the model?*

**Movie Recommendations:** We illustrate how our cover filtration framework could be used to find a sequence of movies to *gently* transition from *Mulan* to *Moulin Rouge*. It would be abrupt to switch directly from one to the other movie, since they belong to completely different genres. We compute stable paths that identify such feasible gentle sequences. Two sequences corresponding to a shortest path and the most stable path are presented in Table 1. See Appendix C.1 for details.

Table 1: Two sequences of movie transitions from *Mulan* to *Moulin Rouge*.

| Shortest Path | Most Stable Path |
|---|---|
| 1. Mulan (1998) | 1. Mulan (1998) |
| 2. Dumbo (1941) | 2. Robin Hood (1973) |
| 3. Sound of Music, The (1965) | 3. Dumbo (1941) |
| 4. Moulin Rouge (2001) | 4. Sound of Music, The (1965) |
| | 5. Gone with the Wind (1939) |
| | 6. Psycho (1960) |
| | 7. High Fidelity (2000) |
| | 8. Moulin Rouge (2001) |

**Explanations in a Machine Learning Model:** With a goal to develop a method of model induction for inspecting a machine learning model, we build a Mapper (Singh et al., 2007) from the predicted probability space of a logistic regression model built from the Fashion-MNIST data set (Xiao et al., 2017). The goal is to understand the model structure by characterizing the relationship between the feature space and the prediction space. We extend the constructed Mapper to be a *Steinhaus Mapper Filtration* and analyze the stable paths in that object.

See Appendix C.2 for details. We look at three regions of shoes (sneaker, ankle boot, and sandals) that are understandably confusing to the machine learning model. We try to elucidate where these confusions arise by studying the stable paths connecting representative nodes for each type of shoe. Figure 11 shows representatives from each vertex in the path for the shortest path and the stable path. In both paths, the vertices start predominately containing sneakers and sneaker-like sandals. They then transition to containing a larger proportion of ankle boots, with all three classes showing higher cut tops or high heels.

Along each path we can see the relationships between nodes change. In the most stable path on the right, we a slow transition from sneaker space to ankle boot space, with some amount of sandals spread throughout. Through the path, shoes from each of the three classes become taller. Near the middle of the path, the images from sneakers and ankle boots are nearly indistinguishable. And earlier in the path, we see how some white strips in the sneakers and boots might easily be confused with negative space in the sandals.

## 7 CONCLUSION

In this work, we established the cover filtration, a new kind of filtration that enables application of TDA to previously inaccessible types of data. We then developed a theory of stable paths in the cover filtration, and provide algorithms for computing the Pareto frontier between short and stable paths. As proof of their utility to real world applications, we show how these two ideas can be applied to the analysis of recommendation systems and of Mapper in the context of explainable machine learning.

The results in this paper suggest many new questions. The application of recommendation systems leaves us curious if the cover filtration along with new results such as the one on predicting links in graphs using persistent homology (Bhatia et al., 2018) could provide methods for answering the main question in recommendation system research: *what item to recommend to the user next?* Proving Conjecture 4.1 is a highly desirable goal.

The applications of cover filtration and stable paths are not limited to ones we highlighted. Another possibility not explored is for sensor networks. Sensor coverage areas are often not uniform balls, and the cover filtration is aptly suited for developing a filtration. In the context of communication networks, stable paths could be interpreted as reliable routes. Yet another direct application could be in finding driving directions that take not only short, but also "easy" routes.

While paths and connected components are most amenable to practical interpretations, could other structures in the cover filtration also suggest insights? What would holes and loops in the cover filtration for recommendation systems mean?

Our application for explanations in machine learning models raises the following question. What are the implications of understanding a path as in Figure 11? How could the predictive model be updated to account for such implications?

We showed that the cover filtration is stable to small changes within the cover. Does this result imply that the persistence diagram of the Steinhaus Mapper filtration is stable with respect to changes in the data, cover parameters, or filter functions?

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

# Appendix

## A    MISSING PROOFS

We restate theorems and lemmas stated without proofs in the main paper, and present their proofs.

**Theorem 2.5.** *The Steinhaus nerve of a cover $\mathcal{U}$ is a filtered simplicial complex.*

*Proof.* This proof makes use of standard set theory results. Let $\mathcal{U}$ be an arbitrary cover of some set $X$ and let $\mathrm{Nrv}_{St}$ be its Steinhaus nerve. We consider $\mathrm{Nrv}_{St}$ as a filtration by assigning as the birth time of simplex $\sigma \in \mathrm{Nrv}_{St}$ its weight $w_\sigma$. To show this is indeed a filtration, we focus on a single simplex $\sigma$ and a face $\tau \preceq \sigma$ to show that the face always appears in the filtration before the simplex.

Suppose $\sigma$ is generated from cover elements $\{U_i\}_{i \in I}$ over some index set $I$. Let a face $\tau \preceq \sigma$ be generated by cover elements indexed by a subset $J \subset I$.

$$\text{The birth time of } \tau \text{ is } d_{St}(\{U_i\}_{i \in J}) = 1 - \frac{\mu(\cap_{i \in J} U_i)}{\mu(\cup_{i \in J} U_i)} \text{ and}$$

$$\text{the birth time of } \sigma \text{ is } d_{St}(\{U_i\}_{i \in I}) = 1 - \frac{\mu(\cap_{i \in I} U_i)}{\mu(\cup_{i \in I} U_i)}.$$

Clearly, with $\{U_i\}_{i \in J} \subset \{U_i\}_{i \in I}$, we have that $\mu(\cap_{i \in J} U_i) \geq \mu(\cap_{i \in I} U_i)$ and $\mu(\cup_{i \in J} U_i) \leq \mu(\cup_{i \in I} U_i)$. It follows then that $d_{St}(\tau) \leq d_{St}(\sigma)$. With $K_\alpha$ denoting the subcomplex that includes all simplices in $\mathrm{Nrv}_{St}$ with birth time at most $\alpha \in [0, 1)$, for any $\alpha, \beta \in [0, 1)$ with $\alpha < \beta$, we have $K_\alpha \subseteq K_\beta$. Hence $\mathrm{Nrv}_{St}(\mathcal{U})$ is a monotonic filtration. $\quad\square$

**Proposition 3.2.** *Let $\mathcal{U}, \mathcal{V},$ and $\mathcal{W}$ be finite covers of a finite set $X$ with equal cardinalities, and let $d_B$ denote the bottleneck distance between any pair of these covers as specified in Definition 3.1. Then $d_B$ is a metric.*

*Proof.* We make the following observations.

1. $d_B(\mathcal{U}, \mathcal{U}) = 0$, as matching each set in the cover to itself gives a distance of $0$ and the smallest cardinality of a symmetric distance is $0$, hence this matching gives the minimum possible symmetric difference. Likewise, if $d_B(\mathcal{U}, \mathcal{V}) = 0$, there is a matching where the symmetric difference between each matched pair has cardinality $0$. Hence the sets are equal for each pair in the matching, and hence $\mathcal{U} = \mathcal{V}$.

2. $d_B(\mathcal{U}, \mathcal{V}) \geq 0$, since $0$ is the greatest lower bound for measures.

3. $d_B(\mathcal{U}, \mathcal{V}) = \min_{M \in \mathcal{M}(\mathcal{U}, \mathcal{V})}\{\max_{(U,V) \in M} \mu(U \Delta V)\} = \min_{M \in \mathcal{M}(\mathcal{V}, \mathcal{U})}\{\max_{(V,U) \in M} \mu(V \Delta U)\} = d_B(\mathcal{V}, \mathcal{U})$.

4. Let $d_B(\mathcal{U}, \mathcal{V}) = \alpha$ and $d_B(\mathcal{V}, \mathcal{W}) = \beta$. Then $U_i$ and its matched set $V_j$ differ on a set $D_1$ with measure at most $\alpha$, and $V_j$ and its matched set $W_k$ differ on a set $D_2$ with measure at most $\beta$. Then $U_i$ and $W_k$ differ on some subset of $D_1 \cup D_2$, so their difference has measure at most $\mu(D_1 \cup D_2) \leq \mu(D_1) + \mu(D_2) \leq \alpha + \beta$. Since this result holds for all $i$, there is a matching between $\mathcal{U}$ and $\mathcal{W}$ with a maximum symmetric difference measure of $\alpha + \beta$. Hence $d_B(\mathcal{U}, \mathcal{W}) \leq \alpha + \beta = d_B(\mathcal{U}, \mathcal{V}) + d_B(\mathcal{V}, \mathcal{W})$.

Hence $d_B$ is a metric with respect to covers of equal cardinality. $\quad\square$

**Corollary 3.4.** *Similar to Lemma 3.3, the greatest increase possible in such a scenario comes from assigning the negative of the total weight to the numerator.*

*Proof.* Let $a, b, c, d, e, f$ be numbers as specified. Then $-c > -e$, and $-c - d = -e - f$, and $|b| > |d|, |b| > |f|$, so $a, b, -c, -d, -e, -f$ fulfill the hypotheses of Lemma 3.3. Then we get that $\frac{a + c}{b - d} < \frac{a + e}{b - f}$. $\quad\square$

**Lemma 4.2.** *The VR filtration completely determines the cover filtration in arbitrary dimensions.*

*Proof.* The intersection of two hyperspheres was derived by Li (2011). The volume of intersection of two hyperspheres of equal radius $R$ in $\mathbb{R}^n$ with centers distance $d$ apart is defined as

$$V_\cap^n(R, d) = \frac{\pi^{n/2}}{\Gamma(\frac{n}{2} + 1)} R^n I_{1-(d/2)/R^2}\left(\frac{n+1}{2}, \frac{1}{2}\right)$$

where $\Gamma$ is the gamma function and $I$ is the regularized incomplete beta function:

$$I_z(a, b) = \frac{\Gamma(a+b) \int_0^z u^{a-1}(1-u)^{b-1} du}{\Gamma(a)\Gamma(b)}.$$

We can reduce this equation to

$$V_\cap^n(R, d) = R^n \pi^{(n-1)/2} \frac{\int_0^{1-(d/2)/R^2} u^{(n-1)/2}(1-u)^{-1/2} du}{\Gamma(\frac{n+1}{2})}.$$

The volume of an $n$-sphere of radius $R$ is

$$V_\circ(R) = \frac{\pi^{n/2}}{\Gamma(\frac{n}{2}+1)} R^n$$

and so the volume of union of two $n$-spheres is

$$V_\cup^n(R, d) = 2V_\circ^n(R) - V_\cap^n(R, d).$$

We then compute the Steinhaus distance with Lebesgue measure of two spheres in $\mathbb{R}^n$ and radius $R$ with Euclidean distance $d$ apart as

$$
\begin{aligned}
d_{St}^n(R, d) &= \frac{2V_\circ^n(R) - 2V_\cap^n(R, d)}{2V_\circ^n(R) - V_\cap^n(R, d)} \\
&= \frac{2\frac{\pi^{n/2}}{\Gamma(\frac{n}{2}+1)} R^n - 2R^n \pi^{(n-1)/2} \frac{\int_0^{1-(d^2/2d)/R^2} u^{(n-1)/2}(1-u)^{-1/2} du}{\Gamma(\frac{n+1}{2})}}{2\frac{\pi^{n/2}}{\Gamma(\frac{n}{2}+1)} R^n - R^n \pi^{(n-1)/2} \frac{\int_0^{1-(d^2/2d)/R^2} u^{(n-1)/2}(1-u)^{-1/2} du}{\Gamma(\frac{n+1}{2})}} \\
&= \frac{2\Gamma(\frac{n+1}{2}) - n\Gamma(\frac{n}{2})\pi^{-1} \int_0^{1-(d/2)/R^2} u^{(n-1)/2}(1-u)^{-1/2} du}{2\Gamma(\frac{n+1}{2}) - \frac{n}{2}\Gamma(\frac{n}{2})\pi^{-1} \int_0^{1-(d/2)/R^2} u^{(n-1)/2}(1-u)^{-1/2} du}.
\end{aligned}
$$

This equation provides a mapping from the birth time of the edge in the Čech filtration to the birth time of the edge in the cover filtration. Once an $n$ and $R$ are chosen, the equation readily reduces, producing the birth times of a simplex in the cover filtration. □

## B EQUIVALENCE

We detail experimental results suggesting that the 1-skeleton of the Steinhaus filtration and the 1-skeleton the Čech filtration are isomorphic (i.e., the Vietoris-Rips filtration). To estimate the area of intersection of 1-spheres, we use Monte Carlo integration with uniform sampling. The first plot in Figure 6 shows the 50 landmark points along with 20,000 points uniformly sampled around the landmarks. The middle plot shows the persistence diagrams of dimension 0 and 1 for the Vietoris-Rips filtration on the landmarks. Finally, we show an approximated Steinhaus filtration on the landmarks, using the balls with radii 0.5 as the covers. We approximate the Steinhaus filtration similar to how the Vietoris-Rips approximates Čech filtration, i.e., by only computing the 1-skeleton of the nerve, and including any higher order simplices for which all faces are already contained in the filtration, taking the maximum birth time of all faces. We note that the two persistence diagrams have only minor differences (only in dimension 1).

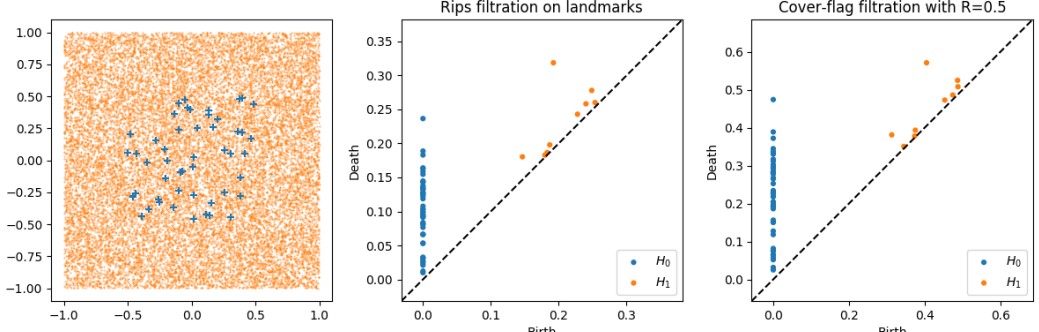

Figure 6: Persistence diagrams for the Vietrois-Rips filtration and the approximate Steinhaus filtration of a set of uniformly sampled points in the plane.

## C APPLICATIONS

### C.1 RECOMMENDATION SYSTEMS

In this application, we apply the cover filtration to a recommendation system data set and employ the stable paths analysis to compute sequences of movies that ease viewers from one title to another title. Our viewpoint on recommendation systems is similar to work of graph-based recommendation systems. This is an active area of research and we believe our new perspective of interpreting such systems as covers and filtrations will yield useful tools for advancing the field. The general approach of graph-based recommendation systems is to model the data as a bipartite graph, with one set of nodes representing the recommendation items and the other set representing the recommendees. We can interpret a bipartite graph as a cover, either with elements being the recommendees covering the items, or elements being the items covering the recommendees.

For an example that we will see more of shortly, suppose you have only ever seen the movie *Mulan* and your partner wants to show you *Moulin Rouge*. It would be jarring to just watch the movie, so your partner might gently build up to *Moulin Rouge* by showing you movies similar to both *Mulan* and *Moulin Rouge*. We compute stable paths that identify such a feasible gentle sequence.

We use the MovieLens-20m data set (Harper & Konstan, 2016). This data set is comprised of 20 million ratings by 138,493 users of 27,278 movies. Often, these types of data sets are interpreted as bipartite graphs. Once we realize that a bipartite graph can be equivalently represented as a covering of one node set with the other, we can apply the cover filtration to build a filtration. In our case, we interpret each movie as a cover element of the users who have rated the movie. To ensure the entire computation runs efficiently on a laptop, and to avoid noise, we remove all movies with less than 10 ratings and then sample 4000 movies at random from the remaining movies.

Figure 7 shows the computed Pareto frontier of stable paths for the case of *Mulan* and *Moulin Rouge*. In Table 1, we show two stable paths that might be chosen. The stable path with length 4 is found after a large drop in instability. As the length and stability must be traded off, we think this would be a decent path to choose if you want to optimize both. The second path shown is the most stable. For readers who have seen the movies in this path, the relationship between each edge is clear, even if one might consider the path a bit on the longer side.

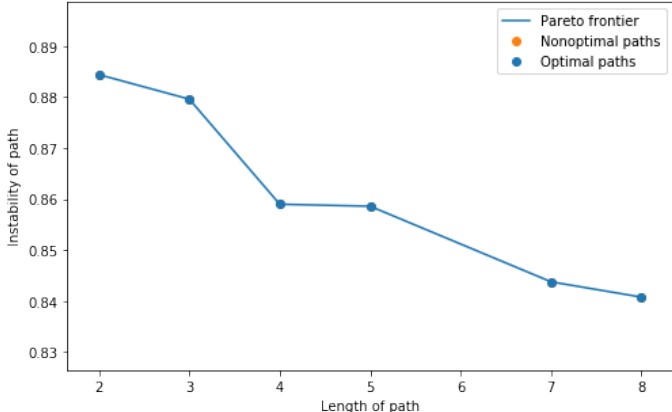

Figure 7: Pareto frontier of stable paths between *Mulan* and *Moulin Rouge*.

Table 1: Two sequences of movie transitions.

| Shortest Path | Most Stable Path |
|---|---|
| 1. Mulan (1998) | 1. Mulan (1998) |
| 2. Dumbo (1941) | 2. Robin Hood (1973) |
| 3. Sound of Music, The (1965) | 3. Dumbo (1941) |
| 4. Moulin Rouge (2001) | 4. Sound of Music, The (1965) |
| | 5. Gone with the Wind (1939) |
| | 6. Psycho (1960) |
| | 7. High Fidelity (2000) |
| | 8. Moulin Rouge (2001) |

### C.2 STEINHAUS MAPPER FILTRATION

As supervised learning has become more powerful, the need for explanations is also grown. We develop a method of model induction for inspecting a machine learning model. The goal is to develop an understanding of the model structure by characterizing the relationship between the feature space and the prediction space. The gleaned understanding can help non-experts make sense of algorithmic decisions and is essential when models are too complex to fully understand in a white-box fashion. The Mapper (Singh et al., 2007) is aptly suited for visualizing this functional structure.

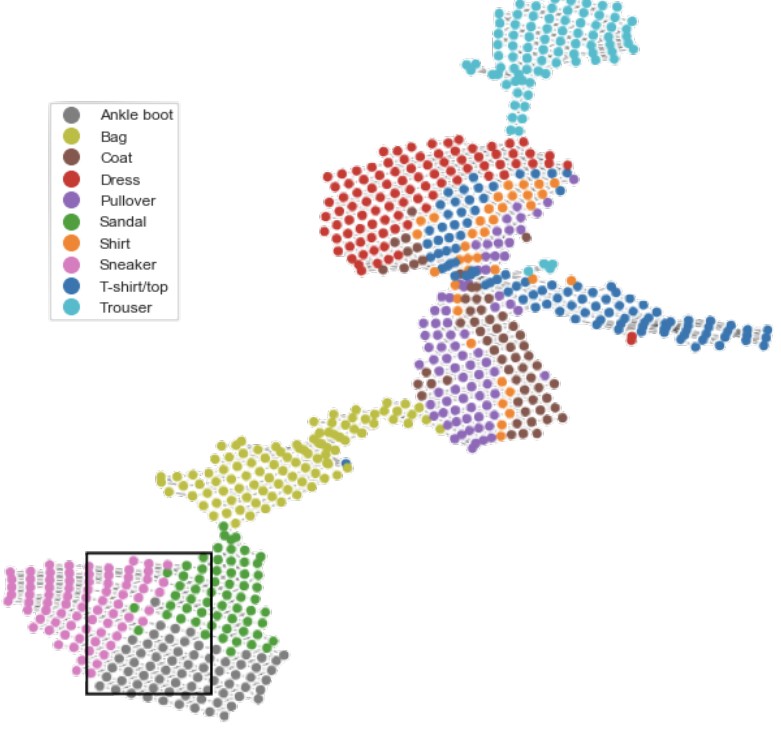

Figure 8: Constructed Mapper from logistic regression model of Fashion-MNIST data set. The window marks the frame of Figure 9

This application is based on the work of using paths in the Mapper to provide explanations for supervised machine learning models (Saul & Arendt, 2018). We build a Mapper from the predicted probability space of a logistic regression model. We then extend the constructed Mapper to be a *Steinhaus Mapper Filtration* and proceed to analyze the stable paths in that object.

Given topological spaces $X, Y$, a function $f : X \to Y$, and a cover of $Y$, we define Mapper to be the nerve of the refined pullback cover of $f(Y)$. A refined cover is one such that each cover element is split into its path-connected components.

**Definition C.1** (Steinhaus Mapper filtration). Given data $X$, a function $f : X \to Y$, and a cover $U$ of $Y$, we define the **Steinhaus Mapper** as the Steinhaus nerve (Definition 2.4) of the refined pullback cover of $f(U)$:

$$\mathrm{Nrv}_{St}(f^*U)\,.$$

By incorporating information about the amount of overlap between the cover produced by the Mapper, our analysis is robust to noise and largely insensitive to the chosen parameters of the Mapper construction.

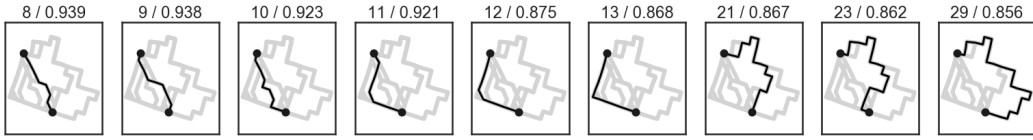

Figure 9: Depiction of stable paths found along Pareto frontier in Figure 10.

Figure 8 shows the Steinhaus Mapper filtration constructed from a logistic regression model built from the Fashion-MNIST data set (Xiao et al., 2017). This data set consists of 70,000 images of clothing items from 10 classes. Each image is 28 pixels. It is widely regarded as a more difficult drop-in replacement for the ubiquitous MNIST handwritten digits data set.

The dimensionality of the data set is first reduced to 100 dimensions using Principal Components Analysis, and then a logistic regression classifier with $l_1$ regularization is trained on the reduced data using 5-fold cross validation on a training set of 60,000 images. The model is evaluated at 93% accuracy on the remaining 10,000 images. We then extract the 10-dimensional predicted probability space and use UMAP (McInnes & Healy, 2018; McInnes et al., 2018) to reduce the space to 2 dimen-

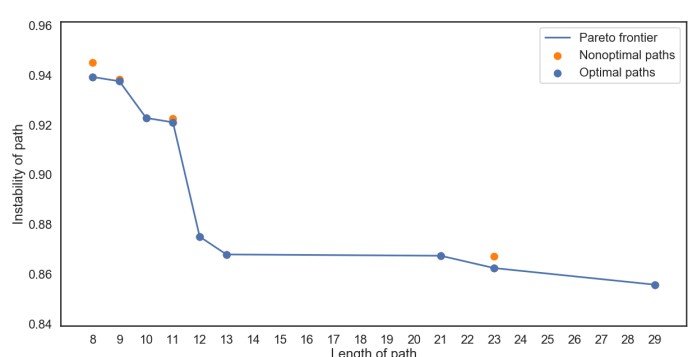

Figure 10: Pareto frontier of stable paths between predominately sneaker vertex and predominately ankle boot vertex.

sions. This 2-dimensional space is taken as the filter function of the Mapper, using a cover consisting of 40 bins along each dimension with 50% overlap between each bin. We explored ranges of values for the number of bins (10–50) and the overlap percentage (20–60%), and observed variabilities in the constructed Mappers. The final choice of parameters (40 bins, 50% overlap) were chosen as the Mappers showed least variability over subsets of values centered at these parameter values. DB-SCAN is used as the clustering algorithm in the refinement step (Ester et al., 1996). KeplerMapper is used for constructing the Mapper (van Veen & Saul, 2017). Finally, the cover is extracted and the Steinhaus Mapper filtration is constructed.

To illustrate the power of the path explanations, we start with two vertices selected from the sneaker and ankle boot regions of the resulting graph. The three regions of shoes (sneaker, ankle boot, and sandals) are understandably confusing to the machine learning model, and we are interested in where these confusions arise. Figure 9 shows the paths associated with the Pareto frontier (Figure 10).

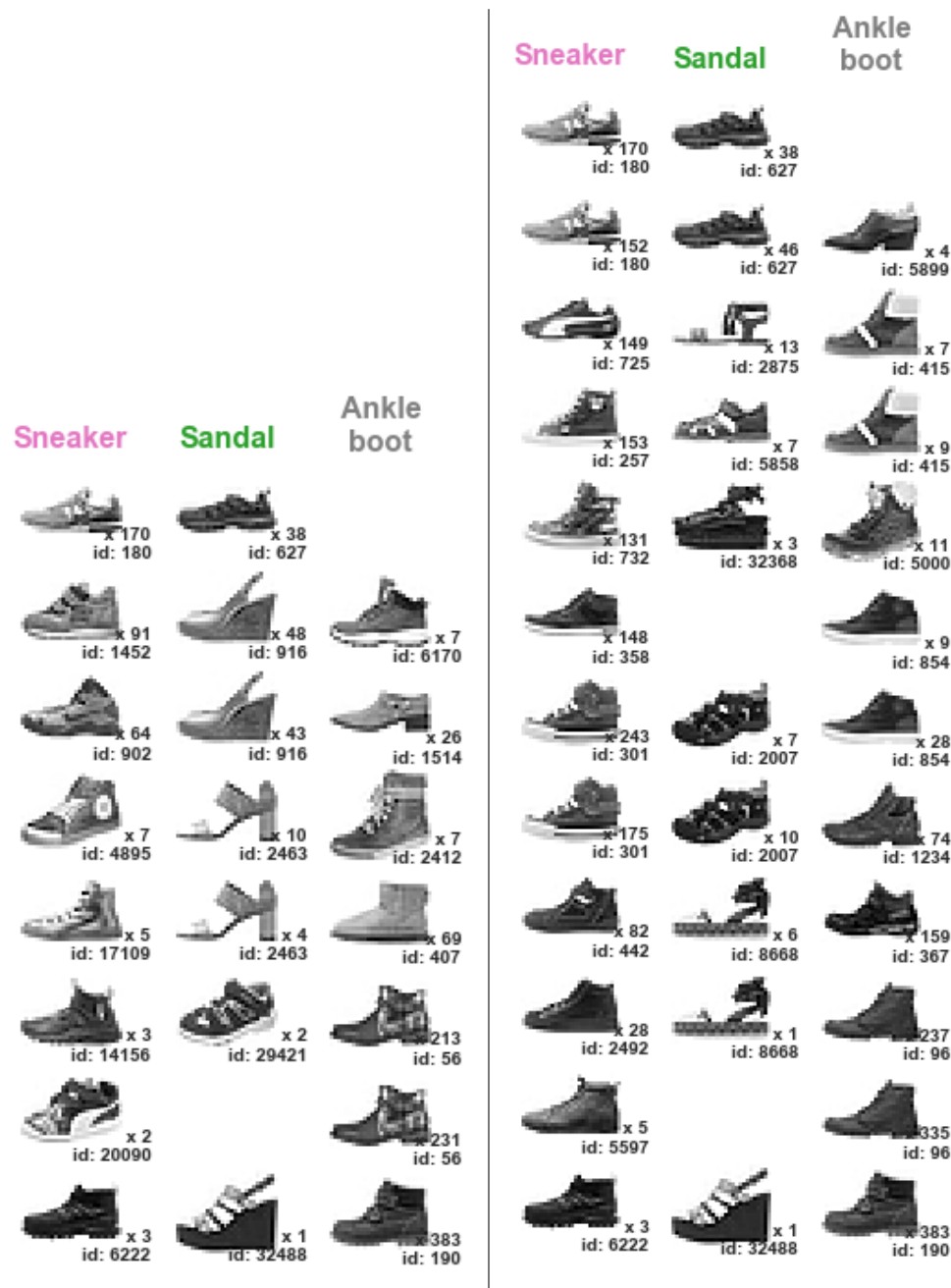

Figure 11: Path visualizations for shortest path (left) and stable path with length 12 (right). Columns in the visualization are based on the class and each row represents a node in the Mapper. We show one representative for each node in each column. Columns with no shoes shown had no representative of that class in the node.

In Figure 10, we show the Pareto frontier between the two chosen vertices. This frontier shows a large decrease in instability value (thus increase in stability) when moving to a path length of 12. As noted in Section 5, paths found after a large increase in stability correspond to highly stable paths, i.e., the path remains the shortest path while sweeping the instability value over a comparatively large range.

Figure 11 shows representatives from each vertex in the path for the shortest path and the stable path with length 12. Each row corresponds to one vertex and the columns show a representative from each class represented in the vertex. Each image shows the multiplicity of that type of shoe in the vertex.

In both paths, the vertices start predominately containing sneakers and sneaker-like sandals. They then transition to containing a larger proportion of ankle boots, with all three classes showing higher cut tops or high heels.

Along each path we can see the relationships between nodes change. In the most stable path on the right, we a slow transition from sneaker space to ankle boot space, with some amount of sandals spread throughout. Through the path, shoes from each of the three classes become taller. Near the middle of the path, the images from sneakers and ankle boots are nearly indistinguishable. And earlier in the path, we see how some white strips in the sneakers and boots might easily be confused with negative space in the sandals.

These two paths provide a holistic representation of how the trained logistic model interprets the data. By exploring these paths, we gain valuable insight into why a model is making a decision. This can help either reinforce our trust in the model or reject the prediction. In either outcome, these explanations can strengthen the results of the predictions by including humans in the loop. Even though the case of predicting clothing types is a low stakes application, this framework is readily applicable to much more important data sets.

## D   NOTE ON COMPLEXITY OF COVER FILTRATIONS

The complexity of constructing the cover filtration is by and large inherited directly from the computational complexity of the nerve. Given a cover $\mathcal{U}$, the nerve could have at most $2^{|\mathcal{U}|} - 1$ simplices and dimension at most $|\mathcal{U}| - 1$ Otter et al. (2017). These bounds are equivalent to the corresponding worst case bounds for VR and Čech complexes.

The work involved for each simplex in constructing $\mathrm{Nrv}_{St}$ includes computing the volume of intersection and volume of union of the elements in the simplex. The complexity of union and intersection operations is largely dependent on the type of data being used. Let $\mathscr{C}_{\mathrm{Unn}}(\mathcal{V})$ and $\mathscr{C}_{\mathrm{Int}}(\mathcal{V})$ be the costs of computing the union and intersection, respectively, of a set of cover elements $\mathcal{V} \subseteq \mathcal{U}$. In the worst case, we have to do $\mathscr{C}_{\mathrm{Unn}}(\mathcal{U}) + \mathscr{C}_{\mathrm{Int}}(\mathcal{U})$ operations per simplex, leading to an overall worst case computational complexity of $(\mathscr{C}_{\mathrm{Unn}}(\mathcal{U}) + \mathscr{C}_{\mathrm{Int}}(\mathcal{U}))(2^{|\mathcal{U}|} - 1)$. For instance, if we assume that a hashing-based dictionary could be produced for each set in $\mathcal{U}$, both $\mathscr{C}_{\mathrm{Unn}}(\mathcal{V})$ and $\mathscr{C}_{\mathrm{Int}}(\mathcal{V})$ will be at most linear in $|\mathcal{V}|$ Bille et al. (2007).

