# OpenReview forum: "Cover Filtration and Stable Paths in the Mapper"
_ICLR.cc/2020/Conference — Reject_

### Official Review · AnonReviewer1 · 2019-10-20
**Official Blind Review #1**

**Rating:** 1

**Review:**

1. Paper summary

This paper develops a novel filtration for the analysis of based on the
idea of *covers* of data sets. The filtration employs the notion of
a generalised Jaccard distance to define stable paths within the nerve
of the cover. These stable paths are then shown to be useful for the
creation of 'gentler' transitions for recommendation systems, as well
as the development of explainable supervised machine learning models.

2. Review summary

I found this paper to be very interesting in terms of its new
perspective on filtrations and its idea of the introducing a
'stability' concept inspired by topological persistence.
However, I recommend rejecting the paper in its current form due to the
following issues:

A. Missing clarity: the paper requires readers to be familiar with the
   Mapper algorithm and glosses over important details. Several results
   that are mentioned as core contributions in the main texts are only
   stated in the supplementary materials.

B. Missing experimental depth: the experiments shown in the paper are
   interesting, but only scratch the surface. To have a convincing case
   study, a more quantitative analysis is required. At present, the
   selection and depiction of results seems ad-hoc. I realise that,
   when the foremost goal is interpretability, providing a quantitative
   analysis is not always easily possible. Nevertheless, each data set
   should be used for more than one case study; ideally, multiple
   results that are 'surprising' can be reported. For example, are
   there paths that do not match our intuition? What do they tell us
   about the data set?

C. Doubts about the technical correctness: the supervised learning
   example requires an elaborate setup (dimensionality reduction,
   UMAP for further reductions, setting up Mapper with additional
   parameters). I am wondering how trustworthy any result obtained
   from such an analysis can be. I would suggest discussing the
   parameter selection process in more detail and ideally providing more
   information about the impact of these choices.

In the following, I will comment on these issues in more detail.

3. Clarity

- Concepts such as 'cover' should be briefly introduced; already the
  abstract presumes that readers are familiar with several TDA concepts

- I would shorten the abstract to improve its flow; the paragraphs 'We
  demonstrate...' and 'For explainable...' could also be added to the
  introduction to improve the exposition of the paper.

- The introduction already starts with TDA concepts; a brief motivation
  would make the paper more accessible

- I disagree with the statement that the existence of a metric on the
  data is an implicit assumption of TDA. My perspective is quite the
  opposite: TDA is rather flexible *because* of its support for
  different metrics; Vietoris--Rips complex calculation, for example,
  only requires a matrix of pairwise distances. Hence, I would rewrite this
  sentence.

- To add to the previous point: the paper itself introduces a new
  metric based on a generalised Jaccard distance. I feel that the
  discussion of 'ill-fit or incomplete' metrics detracts from the
  main message of the paper.

- The first explanation/definition of Mapper in the introduction is
  rather complicated ('nerve of a refined pullback cover'); I would
  suggest rephrasing this.

- The point about the $1$-skeleton warrants more explanation: it is my
  understanding that the paper uses paths that are defined using this
  skeleton as well. My first pass of the paper slightly confused me
  because I figured that the described paths would be generalised over
  high-dimensional simplices as well. I would suggest making this
  'restriction' (it is not a proper restriction because paths can be
  defined for arbitrary simplices, but it is a restriction in terms of
  the dimensionality) or property clear from the beginning.

- The discussion of hypercube covers is repeated multiple times in the
  introduction and the related work section; I would suggest mentioning
  this only once as it does not have to a large bearing on the methods
  described in the paper anyway.

- Discussing stability of the paths/cover in the introduction is
  misleading because these aspects are only discussed in the appendix.
  Same goes for some of the contributions in Section 1.1; the
  conjectures or connection to other filtrations are only mentioned in
  the appendix.

  This needs to be rewritten; I would pick a few contributions that the
  main text can focus on and mention the rest in passing.

  The paper lacks this structure at present and for a conference
  submission, all main contributions should also be a part of the main
  text.

- Stability is meant to reflect a property of a cover; this could be
  clarified by means of extending Figure 1, for example. During my
  first pass of the paper, some of the subsequent definitions were
  lacking a clear motivation. I would suggest to _clearly_ state that
  the goal is to circumvent issues with paths in a cover that only
  depend on a few data points.

- Moreover, it would be interesting to point out to what extent the
  method presented here is the _only_ one capable of doing this. It
  occurred to me that there's a preprint that discusses how to stabilise
  the calculations of persistent homology (with respect to picking the
  _same_ creator simplex for a feature, for example):

    Bendich, Paul et al.
    Stabilizing the unstable output of persistent homology computations
    https://arxiv.org/abs/1512.01700

  It seems that one of the unique features of the method in this paper
  is that the definition of path stability aligns exactly with the other
  stability concept---the paths are thus indeed stable with respect to
  perturbations of the underlying data set.

  Maybe this could be mentioned as an interesting feature.

- The paragraph 'In Section 4.1 [...]' in the related work section is
  somewhat redundant; I would suggest putting it at the beginning of the
  respective experimental section and merging it with the existing
  description there.

- The definition of the Steinhaus distance strikes me as unnecessary
  complex; while it is mathematically pleasing to know that arbitrary
  measures could be used, it seems that this is never exploited anywhere
  in the paper.

- In the interest of clarity, maybe it would make sense to refer to the
  distance as a 'generalised Jaccard distance' instead.

- The proof of Theorem 2.5 could be moved to the appendix. Moreover,
  I think it could be simplified by mentioning that the nerve is
  constructed from the cover and that weights are assigned based on
  the number of intersections; then the main result would follow from
  monotonicity. (the current proof is of course fully correct, I just
  had to rephrase this result in terms of concepts that I found easier
  to grasp)

- Concerning the terminology, I find 'a most stable path' to be somewhat
  confusing at the first read. I now understand what is meant by it, but
  maybe 'highly stable' or 'maximally stable' would be better.

- In Section 3, the 1-skeleton should be explicitly mentioned.

- Moreover, the notion of 'shortest path', which _could_ also employ
  distances, should be distinguished from stable paths; if I understand
  the argumentation correctly, shortest paths are defined in terms of
  the number of edges, while stable paths are defined in terms of the
  weights along those edges. I would suggest spelling this out directly
  to make the concepts non-ambiguous.

- In Definition 3.1, it should be $d_{St}$, not $D_{St}$, I think.

- Definition 3.1 could also be intuitively summarised as 'the largest
  edge weight along a path', right?

- The algorithm in Figure 2 should include some comments that make the
  procedure  more understandable. I found the conceptual leap to Pareto
  frontiers confusing at first; maybe this could be motivated better.

- Figure 3 needs more explanations. In particular the connection to
  persistence could be elucidated---I get that there's an immediate
  connection (see the paper above) but this concept needs at least
  a brief introduction. Moreover, if this connection is relevant, it
  deserves to be elucidated in the paper explicitly.

- Figure 4 needs additional labels; I expect that 'distance/stability'
  are shown in the caption. Is this correct?

  In general, I found the path visualisation not so helpful; maybe it
  would be better to show the full 1-skeleton (or an excerpt) and
  highlight the corresponding paths?

- The language and motivation in Section 4 is somewhat informal; this is
  not an issue for me, but I wanted to mention it as a something that
  could potentially be rephrased.

- The additional application domains should be moved from Section 4 to
  a discussion section.

- The cover generation strategy could be understood more rapidly if
  a small illustration was provided. Do I get it correctly that every
  *cover set* constitutes a film, while the overlap between them is
  based on whether the same user also rated another film?

- The path selection procedure for the experiment in Section 4.1 needs
  to be explained formally. I get that it is akin to selecting a point
  on the 'elbow' of a curve, but this should be briefly explained.

- Figure 5 needs to be referenced more prominently in the text.

- Section 4.2 is glossing over many important details of the definition
  of Mapper. These are absolutely required, though, and any parameters
  selected here should also be analysed to learn whether they affect the
  results.

- I am not sure about the utility of the paths in Figure 7

- Figure 9 needs more explanations; as mentioned below, why not compare
  this path with a geometry-based path in the embedding? Moreover, why
  are there not the same number of images for all classes?

- The conclusion needs to be rewritten; the paper does *not* show
  stability properties in the main text

4. Experiments

- As a general question, I would like to see how stable the
  interpretations of the different sections are. There is always
  a degree of stochasticity when training a neural network, so does
  the output of Mapper stay stable?

- I am not sure about the relevance of experiment in Section 4.1; it
  does not really provide a link to an explainable model for me, but
  instead is more along the lines of 'gentle interpolation in a latent
  space'. I could see that this has its uses, but an evaluation requires
  more than one example. I would be particularly interested in knowing
  about cases in which result is unexpected. Do such cases exist?

- Please comment on the stability of the sampling procedure in Section 4.1

- I understand that Section 4.2 wants to show how algorithms like Mapper can
  act as a middle ground between black-box and white-box models. Is this
  correct? If so, the chosen example is _not_ sufficiently complex to be
  compelling. Would it not be equally appropriate to rewrite this
  experiment in terms of explaining paths in the latent space of
  autoencoders? In this context, I see more of a need for these paths as
  an alternative to paths that purely employ latent space geometry.

- Adding to this, the autoencoder context would it also make possible to
  compare paths generated using Mapper and the proposed as well as the
  paths generated by the model. Such an experiment would be more
  compelling, I think.

- In general, what are the implications of understanding a path as in
  Figure 9? How could the model updated to account for this? If the
  paper were to include an example of how to *fix* a model based on such
  information, it would a highly compelling use case.

- Moreover, would it be possible to detect 'problematic' regions such as
  the one shown in Figure 9 automatically? If so, it would again provide
  a highly compelling example. Otherwise, I think that the
  interpretation of a model depends again on humans and are thus
  restricted based on model complexity.

5. Minor style issues

- Mapper should be capitalised consistently
- I suggest removing the paragraph 'Organization' as it is redundant
- 'mod differences' --> 'modulo differences' (?)
- 'distanceof' --> distance of'
- 'on undirected graph' --> 'on an undirected graph'
- 'Principle component analysis' --> 'Principal component analysis'

**Experience Assessment:**

I have published in this field for several years.

**Review Assessment: Checking Correctness Of Derivations And Theory:**

I carefully checked the derivations and theory.

**Review Assessment: Checking Correctness Of Experiments:**

I carefully checked the experiments.

**Review Assessment: Thoroughness In Paper Reading:**

I read the paper thoroughly.

---

> ### Author Response · Authors · 2019-11-15
> **Responses to Review #1 - Part 1**
>
>
> A. Missing clarity: the paper requires readers to be familiar with the
>    Mapper algorithm and glosses over important details. Several
>    results that are mentioned as core contributions in the main texts
>    are only stated in the supplementary materials.
>
>    -> We have reorganized the paper substantially. We now have
>       stability and equivalence results in the main text, and details
>       of experiments in Appendix
>
>
> B. Missing experimental depth: the experiments shown in the paper are
>    interesting, but only scratch the surface. To have a convincing
>    case study, a more quantitative analysis is required. At present,
>    the selection and depiction of results seems ad-hoc. I realise
>    that, when the foremost goal is interpretability, providing a
>    quantitative analysis is not always easily possible. Nevertheless,
>    each data set should be used for more than one case study; ideally,
>    multiple results that are 'surprising' can be reported. For
>    example, are there paths that do not match our intuition? What do
>    they tell us about the data set?
>
>    -> Please see our responses to specific comments below.
>
>
> C. Doubts about the technical correctness: the supervised learning
>    example requires an elaborate setup (dimensionality reduction, UMAP
>    for further reductions, setting up Mapper with additional
>    parameters). I am wondering how trustworthy any result obtained
>    from such an analysis can be. I would suggest discussing the
>    parameter selection process in more detail and ideally providing
>    more information about the impact of these choices.
>
>    -> Please see our responses to specific comments below.
>
>
> In the following, I will comment on these issues in more detail.
>
> 3. Clarity
>
> i.    Concepts such as 'cover' should be briefly introduced; already the
>       abstract presumes that readers are familiar with several TDA
>       concepts
>
>       -> We have added a 2D illustration of the default Mapper
>          construction in the Introduction, with a figure.
>
>
> ii.   I would shorten the abstract to improve its flow; the paragraphs
>       'We demonstrate...' and 'For explainable...' could also be added
>       to the introduction to improve the exposition of the paper.
>
>       -> Abstract modified and shortened. Those details are presented
>          in the intro (last para under 1.1 Our Contributions).
>
>
> iii.  The introduction already starts with TDA concepts; a brief
>       motivation would make the paper more accessible
>
>       -> We have added more details now.
>
>
> iv.   I disagree with the statement that the existence of a metric on
>       the data is an implicit assumption of TDA. My perspective is
>       quite the opposite: TDA is rather flexible *because* of its
>       support for different metrics; Vietoris--Rips complex
>       calculation, for example, only requires a matrix of pairwise
>       distances. Hence, I would rewrite this sentence.
>
>       -> Rewritten. See response below.
>
>
> v.    To add to the previous point: the paper itself introduces a new
>       metric based on a generalised Jaccard distance. I feel that the
>       discussion of 'ill-fit or incomplete' metrics detracts from the
>       main message of the paper.
>
>       -> Removed the words "ill-fit or incomplete".
>
>
> vi.   The first explanation/definition of Mapper in the introduction is
>       rather complicated ('nerve of a refined pullback cover'); I
>       would suggest rephrasing this.
>
>       ->Changed it to make things more intuitive. Removed the jargon
>         sentence.
>
>
>
> vii.  The point about the 1-skeleton warrants more explanation: it is
>       my understanding that the paper uses paths that are defined
>       using this skeleton as well. My first pass of the paper slightly
>       confused me because I figured that the described paths would be
>       generalised over high-dimensional simplices as well. I would
>       suggest making this 'restriction' (it is not a proper
>       restriction because paths can be defined for arbitrary
>       simplices, but it is a restriction in terms of the
>       dimensionality) or property clear from the beginning.
>
>       ->We state explicitly that we work with 1-skeleton now.
>
>       	But technically, the framework could look at d-chains in the
>       	Mapper for d >= 2. Of course, it will be harder to interpret
>       	such d-chains in the case of real data.
>
>
> viii. The discussion of hypercube covers is repeated multiple times in
>       the introduction and the related work section; I would suggest
>       mentioning this only once as it does not have to a large bearing
>       on the methods described in the paper anyway.
>
>       -> We mention it only once now.

---

> > ### Author Response · Authors · 2019-11-15
> > **Responses to Review #1 - Part 2**
> >
> > ix.   Discussing stability of the paths/cover in the introduction is
> >       misleading because these aspects are only discussed in the
> >       appendix.  Same goes for some of the contributions in Section
> >       1.1; the conjectures or connection to other filtrations are only
> >       mentioned in the appendix.
> >
> >       This needs to be rewritten; I would pick a few contributions
> >       that the main text can focus on and mention the rest in passing.
> >
> >   The paper lacks this structure at present and for a conference
> >   submission, all main contributions should also be a part of the main
> >   text.
> >
> >       ->We have now brought stability into main text, send some experimental
> >         detail to the appendix.
> >
> >
> > x. Stability is meant to reflect a property of a cover; this could be
> >   clarified by means of extending Figure 1, for example. During my
> >   first pass of the paper, some of the subsequent definitions were
> >   lacking a clear motivation. I would suggest to _clearly_ state that
> >   the goal is to circumvent issues with paths in a cover that only
> >   depend on a few data points.
> >
> >       ->Bring discussion of overlaps containing single point into main
> >          text, not just caption to figure 1?
> >
> >
> > xi. Moreover, it would be interesting to point out to what extent the
> >   method presented here is the _only_ one capable of doing this. It
> >   occurred to me that there's a preprint that discusses how to stabilise
> >   the calculations of persistent homology (with respect to picking the
> >   _same_ creator simplex for a feature, for example):
> >
> >     Bendich, Paul et al.
> >     Stabilizing the unstable output of persistent homology computations
> >     https://arxiv.org/abs/1512.01700
> >
> >   It seems that one of the unique features of the method in this paper
> >   is that the definition of path stability aligns exactly with the other
> >   stability concept---the paths are thus indeed stable with respect to
> >   perturbations of the underlying data set.
> >
> >   Maybe this could be mentioned as an interesting feature.
> >
> >   -> Thanks much for pointing us to this paper. We now cite it in the
> >      3rd paragraph in Related Work, and also point out the unique
> >      feature related to stability that out framework provides.
> >
> >
> >
> > xii. The paragraph 'In Section 4.1 [...]' in the related work section is
> >   somewhat redundant; I would suggest putting it at the beginning of the
> >   respective experimental section and merging it with the existing
> >   description there.
> >
> >   ->We moved this para to the start of the Subsection on Rec Sys.
> >
> >
> > xiii. The definition of the Steinhaus distance strikes me as unnecessary
> >   complex; while it is mathematically pleasing to know that arbitrary
> >   measures could be used, it seems that this is never exploited anywhere
> >   in the paper.
> >
> >   -> We now present results on equivalence of our construction to
> >      Cech/VR filtrations in the main text. The more general Steinhaus
> >      distance is needed in that context.
> >
> >
> > xiv. In the interest of clarity, maybe it would make sense to refer to the
> >   distance as a 'generalised Jaccard distance' instead.
> >
> >   -> See above. We mention we use only the generalized Jaccard
> >      distance mostly.
> >
> > xv. The proof of Theorem 2.5 could be moved to the appendix. Moreover,
> >   I think it could be simplified by mentioning that the nerve is
> >   constructed from the cover and that weights are assigned based on
> >   the number of intersections; then the main result would follow from
> >   monotonicity. (the current proof is of course fully correct, I just
> >   had to rephrase this result in terms of concepts that I found easier
> >   to grasp)
> >
> >      ->The proof is now moved to Appendix.
> >
> >
> > xvi. Concerning the terminology, I find 'a most stable path' to be somewhat
> >   confusing at the first read. I now understand what is meant by it, but
> >   maybe 'highly stable' or 'maximally stable' would be better.
> >
> >      ->Replaced 'most stable path' with 'maximally stable path'
> >
> >
> > xvii. In Section 3, the 1-skeleton should be explicitly mentioned.
> >
> >      ->We explicitly mention 1-skeleton now.
> >
> > xviii. Moreover, the notion of 'shortest path', which _could_ also employ
> >   distances, should be distinguished from stable paths; if I understand
> >   the argumentation correctly, shortest paths are defined in terms of
> >   the number of edges, while stable paths are defined in terms of the
> >   weights along those edges. I would suggest spelling this out directly
> >   to make the concepts non-ambiguous.
> >
> >   -> Added clarification at start of the Section to this effect.
> >
> >
> > ixx. In Definition 3.1, it should be d_st, not D_st, I think.
> >
> >      ->Replaced. Also in Figure 2.
> >
> > xx. Definition 3.1 could also be intuitively summarised as 'the largest
> >   edge weight along a path', right?
> >
> >      -> Yes. We've added a sentence to this effect right after the
> >         Definition (Definition 5.1 now).

---

> > > ### Author Response · Authors · 2019-11-15
> > > **Responses to Review #1 - Part 3**
> > >
> > >
> > > xxi. The algorithm in Figure 2 should include some comments that make the
> > >   procedure  more understandable. I found the conceptual leap to Pareto
> > >   frontiers confusing at first; maybe this could be motivated better.
> > >
> > >      -> We present the motivation and details in the para to the right
> > >         of Figure 2 (the algorithm).
> > >
> > >
> > >
> > > xxii. Figure 3 needs more explanations. In particular the connection to
> > >   persistence could be elucidated---I get that there's an immediate
> > >   connection (see the paper above) but this concept needs at least
> > >   a brief introduction. Moreover, if this connection is relevant, it
> > >   deserves to be elucidated in the paper explicitly.
> > >
> > >       -> We are not making any strong claims about the direct
> > >          correspondence to persistence. It is only a superficial
> > >          motivation, and hence we changed the language to say
> > >          "...similar in a loose sense to computing persistent
> > >          homology" (at the end of first para in this Section).
> > >
> > >
> > > xxiii. Figure 4 needs additional labels; I expect that 'distance/stability'
> > >   are shown in the caption. Is this correct?
> > >
> > >   In general, I found the path visualisation not so helpful; maybe it
> > >   would be better to show the full 1-skeleton (or an excerpt) and
> > >   highlight the corresponding paths?
> > >
> > >   -> We added "Length/\rho shown on top" to the caption.
> > >
> > >      We highlight the portion of Mapper in Figure 8 on which the
> > >      stable/short paths from the corresponding Pareto frontier (Fig
> > >      10) are shown.
> > >
> > >
> > > xxiv. The language and motivation in Section 4 is somewhat informal; this is
> > >   not an issue for me, but I wanted to mention it as a something that
> > >   could potentially be rephrased.
> > >
> > >
> > >
> > > xxv. The additional application domains should be moved from Section 4 to
> > >   a discussion section.
> > >
> > >    ->  Moved to Conclusion.
> > >
> > >
> > > xxvi. The cover generation strategy could be understood more rapidly if
> > >   a small illustration was provided. Do I get it correctly that every
> > >   *cover set* constitutes a film, while the overlap between them is
> > >   based on whether the same user also rated another film?
> > >
> > >    -> That is correct.
> > >
> > >
> > >
> > > xxvii. The path selection procedure for the experiment in Section 4.1 needs
> > >   to be explained formally. I get that it is akin to selecting a point
> > >   on the 'elbow' of a curve, but this should be briefly explained.
> > >
> > >   -> We explain the path selection in the paragraph to the left of
> > >      Figure 7 (the Pareto frontier figure in this Section). There is a
> > >      marked increase in stability from the 3-edge path to 4-edge path,
> > >      and hence we choose the 4-edge path as the shortest path.
> > >
> > >
> > >
> > > xxviii. Figure 5 needs to be referenced more prominently in the text.
> > >
> > >    -> Moved to Appendix now, and mentioned in text right next to the
> > >       Figure.
> > >
> > >
> > > ixxx. Section 4.2 is glossing over many important details of the definition
> > >   of Mapper. These are absolutely required, though, and any parameters
> > >   selected here should also be analysed to learn whether they affect the
> > >   results.
> > >
> > >    -> We have added some details about how we chose the # bins and
> > >       overlap % for the Mapper (below Figure 10).
> > >
> > > xxx. I am not sure about the utility of the paths in Figure 7
> > >
> > >    -> The Figure illustrates how length (in # edges) and stability get
> > >       traded off within the window highlighted in the Mapper in Figure
> > >       7 (old Figure 6).
> > >
> > > xxxi. Figure 9 needs more explanations; as mentioned below, why not compare
> > >   this path with a geometry-based path in the embedding? Moreover, why
> > >   are there not the same number of images for all classes?
> > >
> > >    -> The caption has been expanded. In particular, we added that "
> > >       Columns with no shoes shown had no representative of that class
> > >       in the node."
> > >
> > >
> > > xxxii. The conclusion needs to be rewritten; the paper does *not* show
> > >   stability properties in the main text
> > >
> > >    -> Stability is presented in main text now.

---

> > > > ### Author Response · Authors · 2019-11-15
> > > > **Responses to Review #1 - Part 4**
> > > >
> > > >
> > > > 4. Experiments
> > > >
> > > > i. As a general question, I would like to see how stable the
> > > >   interpretations of the different sections are. There is always
> > > >   a degree of stochasticity when training a neural network, so does
> > > >   the output of Mapper stay stable?
> > > >
> > > >     -> We posit that the theoretical stability guarantees for our
> > > >        framework provides some level of stability to such variations.
> > > >
> > > > ii. I am not sure about the relevance of experiment in Section 4.1; it
> > > >   does not really provide a link to an explainable model for me, but
> > > >   instead is more along the lines of 'gentle interpolation in a latent
> > > >   space'. I could see that this has its uses, but an evaluation requires
> > > >   more than one example. I would be particularly interested in knowing
> > > >   about cases in which result is unexpected. Do such cases exist?
> > > >
> > > >     -> We realize it is tricky to evaluate such recommendations, since
> > > >        there is no "ground truth" to compare. We highlighted the
> > > >        Mulan->Moulin Rouge example since it presented a highly
> > > >        illustrative instance for the kind of paths we are looking
> > > >        for. We explored a few other interesting cases, but have not
> > > >        presented details due to space limitations.
> > > >
> > > >
> > > > iii. Please comment on the stability of the sampling procedure in Section 4.1
> > > >
> > > >     -> The sampling was performed mostly to ensure the overall
> > > >        computational pipeline runs on a typical laptop in good
> > > >        time. There is nothing suggesting outcomes would be drastically
> > > >        different if we try the pipeline on much larger samples, or
> > > >        even the whole dataset.
> > > >
> > > >        Once we received this round of reviews, we considered setting
> > > >        up a run for the whole data set on AWS. But we were not able to
> > > >        finish the experiment on time for the rebuttal. We plan to
> > > >        pursue this experiment in the near future, though.
> > > >
> > > >
> > > >
> > > > iv. I understand that Section 4.2 wants to show how algorithms like Mapper can
> > > >   act as a middle ground between black-box and white-box models. Is this
> > > >   correct? If so, the chosen example is _not_ sufficiently complex to be
> > > >   compelling. Would it not be equally appropriate to rewrite this
> > > >   experiment in terms of explaining paths in the latent space of
> > > >   autoencoders? In this context, I see more of a need for these paths as
> > > >   an alternative to paths that purely employ latent space geometry.
> > > >
> > > >     -> Thanks for the suggestion. We will explore this in future work.
> > > >
> > > >
> > > > v. Adding to this, the autoencoder context would it also make possible to
> > > >   compare paths generated using Mapper and the proposed as well as the
> > > >   paths generated by the model. Such an experiment would be more
> > > >   compelling, I think.
> > > >
> > > >     -> Thanks for the suggestion. We will explore this in future work.
> > > >
> > > >
> > > > vi. In general, what are the implications of understanding a path as in
> > > >   Figure 9? How could the model updated to account for this? If the
> > > >   paper were to include an example of how to *fix* a model based on such
> > > >   information, it would a highly compelling use case.
> > > >
> > > >     -> Again, great idea. Will explore in future work.
> > > >        We have listed this problem in our Conclusions.
> > > >
> > > > 5. Minor style issues
> > > >
> > > > i. Mapper should be capitalised consistently
> > > >
> > > >    ->Replaced all instances of 'mapper' with 'Mapper'
> > > >
> > > > ii. I suggest removing the paragraph 'Organization' as it is redundant
> > > >
> > > >    ->Remove (Done)
> > > >
> > > > iii. 'mod differences' --> 'modulo differences' (?)
> > > >
> > > >    ->Replace (Done)
> > > >
> > > > iv. 'distanceof' --> distance of'
> > > >
> > > >    ->Replace (Done)
> > > >
> > > > v. 'on undirected graph' --> 'on an undirected graph'
> > > >
> > > >    ->Replace (Done)
> > > >
> > > > vi. 'Principle component analysis' --> 'Principal component analysis'
> > > >
> > > >    ->Replace (Done)

---

> > > > > ### Comment · AnonReviewer1 · 2019-11-15
> > > > > **Thanks for the rebuttal**
> > > > >
> > > > > Thank you very much for the detailed rebuttal! Could you please add some more information (as a comment) on the sampling procedure & the dimensionality reduction for the experiments? This is to some extent already addressed in your comments above, but I did not find anything in the paper/comments concerning the stability of the parameter choices. The whole pipeline for Fashion-MNIST still strikes me as very involved, and I think a more in-depth discussion of parameter choices would absolutely be required here.

---

> > > > > > ### Author Response · Authors · 2019-11-15
> > > > > > **FashionMNIST Experiment details**
> > > > > >
> > > > > > Here are some more details for the paragraph next to Figure 10 (in Page 18, part of Appendix) in the paper:
> > > > > >
> > > > > >
> > > > > >     We initially train a model using a very standard and naive approach. First, we reduce the dimensionality of the images to 100 using Principal Components Analysis and use this 100 dimensional space to train a LR model with L1 regularization on the FashionMNIST designated train set of 60k images.
> > > > > >
> > > > > >
> > > > > >     This naive approach is satisfactory to demonstrate the use of the Mapper filtration. It provides us a mapping from a 100 dimensional space to a 10 dimensional space.
> > > > > >
> > > > > >
> > > > > >     Unfortunately, Mapper is unable to operate on a lens of 10 dimensions because of the curse of dimensionality. For this reason, we must reduce this space to ~2 dimensions.  Here, we opt for the dimensionality reduction method de jour UMAP as it preserves some local connectivity in the data set. Any other density based dimensionality reduction would be suitable here as they tend to maintain local relationships of the data, which is most important for preserving when exploring topology. The parameters for UMAP are taken as the default.
> > > > > >
> > > > > >
> > > > > >     For this example, we chose 40 bins with a 50% overlap.  The number of bins is best chosen to help with the visualization density of each node and should be scaled to the number of observations being plotted. In two dimensions, this results in 40*40= 1600 bins and expected 37.5 observations per bin.  This order of magnitude is digestible when viewing the Mapper. As for the overlap, we opt for a larger overlap than is usually used in the literature. This is because we can leverage the stability of paths in later steps to filter out noise resulting in the over connected from high overlap.
> > > > > >
> > > > > >
> > > > > >     Finally, for the choice of clustering algorithm, we opt for DBSCAN as a default as it handles the density of data well. The parameters are taken as the default.
> > > > > >
> > > > > >
> > > > > > Please let us know if we could provide any more specific details.

---

### Official Review · AnonReviewer3 · 2019-11-02
**Official Blind Review #3**

**Rating:** 6

**Review:**

The paper presents an interesting filtration method to find staple maps, which proves effective in the recommendation system and explainable machine learning. The new framework is built upon a new concept termed cover filtration based on the generalized Steinhaus distance derived from Jaccard distance. A staple path discovery mechanism is then developed using this filtration based on persistent homology. Experiments on Movielense and FashionMNIST has quantitatively and qualitatively demonstrated the effectiveness of the new model. It also showcases the explainable factors with visualized samples from FashionMNIST dataset. In addition, theoretical discussion in the appendix makes the theory part of this work solid and convincing.

* The method introduced in this paper is intuitive and demonstrates with meaningful outcomes.

* The method itself is built upon the SOTA method “Mapper”. Authors may want to clearly state the difference between the new method and Mapper in writing.

* The overall structure of this paper needs some improvement. Authors attempt to introduce a new distance metric as an intermediate level representation. This is not very clear at the beginning; instead, most of the introduction is related to TDA and mapper, which may confuse people not from this particular field.

* What is the meaning of D_St under definition 3.1? Is this the same as d_St or not?

* It seems the restrictions in the experiments are strong, which can be found from a few settings in the experiments. For example, the authors claim that “To reduce computational expenses and noise, we remove all movies with less than 10 ratings and then sample 4000 movies at random from the remaining movies.” It seems the model may not work well in a more general case...

* The pre-processing steps for filtration cover seem verbose on FashionMNIST “The model is evaluated at 93% accuracy on the remaining 10,000 images. We then extract the 10-dimensional predicted probability space and use UMAP (McInnes & Healy, 2018) to reduce the space to 2 dimensions. This 2-dimensional space is taken as the filter function of the Mapper, using a cover consisting of 40 bins along each dimension with 50% overlap between each bin.” Not sure if this method will work well without fine-tuning or feature selection like this.


**Experience Assessment:**

I have read many papers in this area.

**Review Assessment: Checking Correctness Of Derivations And Theory:**

I assessed the sensibility of the derivations and theory.

**Review Assessment: Checking Correctness Of Experiments:**

I assessed the sensibility of the experiments.

**Review Assessment: Thoroughness In Paper Reading:**

I read the paper at least twice and used my best judgement in assessing the paper.

---

> ### Author Response · Authors · 2019-11-15
> **Responses to Review #3**
>
>
> 2. The method itself is built upon the SOTA method “Mapper”. Authors
>    may want to clearly state the difference between the new method and
>    Mapper in writing.
>
>    -> We introduce Mapper using a picture in the Introduction now.
>
>
> 3. The overall structure of this paper needs some improvement. Authors
>    attempt to introduce a new distance metric as an intermediate level
>    representation. This is not very clear at the beginning; instead,
>    most of the introduction is related to TDA and mapper, which may
>    confuse people not from this particular field.
>
>    -> We have comprehensively restructured the paper as detailed here
>       and in the our responses to Review #1.
>
>
>
> 4. What is the meaning of D_St under definition 3.1? Is this the same
>    as d_St or not?
>
>    -> Changed  D_st to d_st in def 3.1
>
>
> 5. It seems the restrictions in the experiments are strong, which can
>    be found from a few settings in the experiments. For example, the
>    authors claim that “To reduce computational expenses and noise, we
>    remove all movies with less than 10 ratings and then sample 4000
>    movies at random from the remaining movies.” It seems the model may
>    not work well in a more general case...
>
>    -> Please see response to related comment from Review #1.
>
>
>
> 6. The pre-processing steps for filtration cover seem verbose on
>    FashionMNIST “The model is evaluated at 93% accuracy on the
>    remaining 10,000 images. We then extract the 10-dimensional
>    predicted probability space and use UMAP (McInnes & Healy, 2018) to
>    reduce the space to 2 dimensions. This 2-dimensional space is taken
>    as the filter function of the Mapper, using a cover consisting of
>    40 bins along each dimension with 50% overlap between each bin.”
>    Not sure if this method will work well without fine-tuning or
>    feature selection like this.
>
>    -> Please see response to related comment from Review #1.

---

### Decision · Program_Chairs · 2019-12-19

**Decision:**

Reject

**Comment:**

The paper proposes a filtration based on the covers of data sets and demonstrates its effectiveness in recommendation systems and explainable machine learning. The paper is theory focused, and the discussion was mainly centered around one very detailed and thorough review. The main concerns raised in the reviews and reiterated at the end of the rebuttal cycle was lack of clarity, relatively incremental contribution, and limited experimental evaluation. Due to my limited knowledge of this particular field, I base my recommendation mostly on R1's assessment and recommend rejecting this submission.